# Biological Activities of *Mikania glomerata* and *Mikania laevigata*: A Scoping Review and Evidence Gap Mapping

**DOI:** 10.3390/ph18040552

**Published:** 2025-04-09

**Authors:** Thaís Pelegrin Garcia, Daniela Gorski, Alexandre de Fátima Cobre, Raul Edison Luna Lazo, Gustavo Bertol, Luana Mota Ferreira, Roberto Pontarolo

**Affiliations:** 1Postgraduate Program in Pharmaceutical Sciences, Universidade Federal do Paraná, Curitiba 80020-300, Paraná, Brazil; thais.garcia@ufpr.br (T.P.G.); danielagorski@ufpr.br (D.G.); alexandrecobre@gmail.com (A.d.F.C.); raulluna@ufpr.br (R.E.L.L.); luanamota@ufpr.br (L.M.F.); 2Dall PhytoLab S.A., Curitiba 82540-040, Paraná, Brazil; gubertol@yahoo.com.br; 3Department of Pharmacy, Universidade Federal do Paraná, Curitiba 80020-300, Paraná, Brazil

**Keywords:** guaco, *Mikania glomerata*, *Mikania laevigata*, phytotherapy, plants, medicinal, anti-inflammatory agents

## Abstract

**Background/Objectives**: The species *Mikania glomerata* and *Mikania laevigata* are commonly referred to as guaco. Their preparations are used in traditional Brazilian medicine, mainly to address respiratory conditions affecting the upper airways. Considering the wide popular use of this species, the present study aims to survey the biological activities of guaco that have already been proven in the literature and to generate an evidence gap map for these biological activities. **Methods**: A scoping review was conducted using the electronic databases PubMed, Scopus, and Web of Science (7 October 2024), which included all studies that have evaluated the biological activities of the leaves of the *M. glomerata* or *M. laevigata* species. **Results**: A total of 57 studies (31 assessed only *M. glomerata*, 17 assessed only *M. laevigata*, and 9 assessed both species) evaluating 38 different biological activities demonstrated that preclinical studies reported 23 biological activities for *M. glomerata* and 24 for *M. laevigata*. The most extensively researched activity for both species is their anti-inflammatory properties, which have been associated with their efficacy in treating bronchoconstriction and their popular uses as an antiophidic agent. The gap map illustrates the lack of evidence to support the biological activity of these species, which may explain some of their popular uses, such as their use as expectorants, antipyretics, for arthritis, rheumatism, neuralgia, and as an antisyphilitic. **Conclusions**: Considering these findings, there is a clear need for further studies to evaluate the activity of these species for these purposes, mainly through clinical studies.

## 1. Introduction

*Mikania glomerata* Spreng. and *Mikania laevigata* Sch.Bip. ex Baker, commonly known as guaco, are important medicinal plants native to South America, particularly Brazil. They are often used interchangeably, sometimes without a clear distinction between the two species [1]. The morphoanatomical characteristics of the leaves, petioles, and stems are highly similar, which presents a significant challenge in differentiating them based on these criteria alone [2,3]. 

With regard to the phytochemical profile, there are notable differences between the species. Regarding the presence of coumarin in the leaves, this is more pronounced in *M. laevigata*, whereas it may be absent or present in a low concentration in *M. glomerata* [2,3]. Additionally, *M. glomerata* exhibits higher concentrations of chlorogenic and dicaffeoylquinic acids [2]. The compounds present in the extracts of the leaves of these species, including coumarins, terpenes, kaurenoic acid, aldehydes, and organic esters, have been associated with a range of pharmacological activities [4]. For example, coumarin has been associated with anti-cancer, antioxidant, and anti-inflammatory activities [5].

In Brazil, the leaves of both species are widely used in folk medicine, mainly in the form of infusions to treat colds, flu, and respiratory problems [6]. In addition, Brazilian Indigenous peoples traditionally use guaco to treat snakebites, arthritis, and various inflammatory conditions such as rheumatism, enteritis, ulcers, and fever [1]. In 2007, *M. glomerata* leaf syrup was included in the Reference List of Complementary Medicines and Supplements of the Brazilian Pharmaceutical Assistance and is used, primarily, as a bronchodilator and expectorant [4]. The World Health Organization (WHO) actively promotes strategies to integrate traditional and complementary medicine, including self-care services, into national health systems, highlighting the importance of medicinal plants in health care [7].

Given the extensive utilization of these plants and the lack of systematic reviews that aim to synthesize the evidence pertaining to their biological activities, this scoping review aims to assess the risk of bias of the included studies and map the existing evidence on the biological activity of the extracts from the leaves of *M. glomerata* and *M. laevigata*.

## 2. Results

The search strategy identified 1660 unique records, of which 1564 were excluded during the screening process, leaving 96 that were included for eligibility assessment. Of these, 4 were not retrieved, and 35 were excluded (see the complete list of reasons for exclusion in Appendix A). Ultimately, 57 records were included in the analysis. No additional studies were found through manual research (Figure 1) [6,8,9,10,11,12,13,14,15,16,17,18,19,20,21,22,23,24,25,26,27,28,29,30,31,32,33,34,35,36,37,38,39,40,41,42,43,44,45,46,47,48,49,50,51,52,53,54,55,56,57,58,59,60,61,62,63].

Among these 57 studies published between 1991 and 2019, 31 assessed only *M. glomerata*, 17 assessed only *M. laevigata*, and 9 assessed both species. Seven studies did not indicate the place of origin of the plant used to prepare the extract (Appendix A). The remaining studies originated from Brazil, with the majority (*n* = 35 studies) deriving from the Atlantic Forest region. The most frequently employed extraction method was maceration (*n* = 30), while the most utilized solvent was ethanol (*n* = 38). The majority were conducted using in vivo (*n* = 21) or in vitro (*n* = 24) assays only, and some were conducted using both in vivo and in vitro (*n* = 5). Three studies were conducted in ex vivo models only; one in an ex vivo and in vitro model; and one in vitro, ex vivo, and in vivo. Two randomized clinical trials were also retrieved [14,36]. One evaluated the safety of *M. glomerata* and *M. laevigata* extracts in an open-label study, measuring the occurrence of adverse effects, changes in blood parameters, and blood pressure [14]. The second randomized clinical trial (RCT) evaluated the bronchodilator effect of *M. glomerata* extract in asthmatic patients in a double-blind study compared with standard treatment (salbutamol) and a placebo [36]. In total, 34 distinct activities were assessed.

The risk of bias assessment using the Cochrane Risk of Bias tool for randomized trials (ROB2) showed that there was a low risk of bias for the RCTs assessing bronchodilator efficacy and for the outcomes of blood pressure and blood parameters in the RCTs assessing toxicity (Appendix A). There were some concerns about the outcome of adverse effects due to the unblinded nature of the study. When assessing bias in animal model studies using the SYRCLE tool, all studies had a moderate-to-high risk of bias, mainly due to a lack of information on the blinding of investigators and assessors and on the randomization and evaluation process (Appendix A).

### 2.1. Mikania glomerata

The most studied aspect of *M. glomerata* has been its anti-inflammatory activity, with a total of 11 studies conducted [16,18,25,27,32,33,35,48,51,54] (Table 1). Of these, two demonstrated anti-inflammatory effects in in vitro assays (peripheral blood mononuclear cells and screening kits) [18,48] and in 7 in in vivo models using Wistar rats [8,25,26,32,33,35,51,54]. Five of these models examined edema caused by envenomation from *Bothropoides jararaca* (*n* = 3 studies) [8,25,51,54] and *Crotalus durissus* (*n* = 1) [33]. Other models included pleural edema, lung inflammation caused by coal dust, paw edema, peritoneal inflammation, and pleurisy [26,32,35]. This anti-inflammatory activity was correlated with the presence of coumarin in the extracts [27,51], with the primary mechanism of action being the inhibition of phospholipase A2 [51,54]. Additionally, it was associated with the dual inhibition of cyclooxygenase-1 and 5-lipoxygenase [18], prostaglandin synthesis [26], leukocyte influx [32], and the blockade of lymphocyte stimulation [48]. Among the studies that reported an absence of this activity, one was conducted in vivo in rats with *Bothrops* and *Crotalus durissus* terrifilis envenomation [44], while another was in vitro, using dystrophic primary skeletal muscle cell cultures [16] (Appendix A).

In addition to anti-inflammatory activity against snake venom, this species also showed protective activity against other symptoms associated with envenomation, such as anti-hemorrhagic (*n* = 2 studies) [44,51], anti-sedative (*n* = 1) [33], and analgesic (*n* = 1) effects [54] (Table 1). This species has also been shown to be anxiolytic, acting to increase GABA levels and reduce glutamate and aspartate concentrations in the hippocampus of mice [57] (Table 1). Bronchodilator activity was reported in an ex vivo study involving human bronchi and guinea pig tracheae [25], although this activity was not observed in a clinical study in asthmatic patients [36] (Table 1).

Antimicrobial activity was evaluated in six in vitro studies [17,39,42,49,50,61], with five showing that *M. glomerata* extracts exhibited activity against 21 different bacteria [39,42,49,50,61], which was correlated with the presence of diterpenes [49,50], kaurenoic acids [61], coumarin [42,61], and cupressenic acid [61] (Table 1). Yatusuda et al. reported that diterpenes and coumarin were the major compounds present in the extract used, and they correlated the antimicrobial activity against *M. streptococci*. However, the authors did not describe the mechanism of action of these compounds [61]. Similarly, the study by Lessa et al. associated coumarin with antimicrobial activity. In their study, a coumarin solution was tested against *M. streptococci*; this exhibited a minimal inhibitory concentration (MIC) of 15 µg/mL [42]. In addition, an ethanol extract mouthwash containing *M. glomerata* presents a low concentration of coumarin, being lowest at the limit of quantification of HPLC; however, this mouthwash showed an MIC of 125 µg/mL. These findings suggest that, although coumarin is a key contributor to the antimicrobial activity, the efficacy of the extract may also be influenced by other bioactive compounds or synergistic effects within the extract.

Two studies reported no activity against Candida albicans [30,39], and one study found no activity of the ethanolic extract against cultures of *Escherichia coli*, *Pseudomonas aeruginosa*, *Bacillus subtilis*, *Staphylococcus aureus*, *Candida krusei*, *Candida parapsilosis*, and *Candida tropicalis* [39] (Appendix A). However, one study demonstrated the potential of methanolic extract against *E. coli* and *P. aeruginosa* [17]. As well as demonstrating efficacy in Streptococcus mutans cultures, two studies have shown the potential of this species to treat biofilms of this bacterium [42,49], one of which was carried out on biofilms from children’s toothbrushes containing resistant strains of this bacterium [42].

Antiviral activity was noted in two studies (against human herpes simplex virus types 1 and 2, Suid alphaherpesvirus 1, and Bovine Herpesvirus type-1) [40,59]. In the study conducted by Silva et al., the coumarin showed activity against *herpes simplex virus* types 1, but not against *herpes simplex virus* types 2 [59]. Thus, the authors correlated the activity against herpes simplex virus types 2 with the synergy of the coumarin present in the extract with the other compounds.

Molluscicidal activity was observed in the study conducted by Souza et al., against *Subulina octona* [26]. This activity was associated with the presence of saponins, tannins, and flavonoids. The authors suggest that the mechanism of action of saponins is based on cell lysis and complexation with steroids. For the tannins and flavonoids, the proposed mechanism involves protein complex formation and the inhibition of the detoxification system. However, this suggestion is based on the foundation of the literature; this hypothesis is not a test for the saponins, tannins and flavonoids present in the *M. glomerata*.

Anthelmintic activity was observed in a study against *Toxocara canis* and *Ancylostoma caninum* [62]. Zamprogno et al. correlated this activity with the presence of terpenes, which act by inhibiting the embryogenesis of eggs [62]. Nematocidal activity was evaluated in one study against *Pratylenchus jaehni* and *Pratylenchus zeae* [63], and antiprotozoal activity was evaluated in one study against *Leishmania amazonensis* [43], but the mechanism of action of this activity was not elucidated (Table 1). The extract was also tested against *Herpetomonas samuelpessoai* [11] and *Trypanosoma cruzi* [43], showing no activity against these organisms (Appendix A).

This species also exhibited antioxidant (*n* = 3 studies) [16,19,57], anti-allergic (*n* = 1) [32], anti-antiproliferative (*n* = 1) [24], antispasmodic (*n* = 1) [8], hemolytic (*n* = 1) [43], hepatoprotective (*n* = 1) [19], and vasodilatory activity (*n* = 1) [25] (Table 1). In contrast to the diverse pharmacological activities identified, some areas in which *M. glomerata* did not demonstrate potential include anticoagulant activity (*n* = 1) [14], as a hemolytic (*n* = 1) [44], as a sedative [57], as a muscle relaxant [57], and reproductive toxicity (*n* = 3) [22,23,28,55] (Appendix A).

Six studies evaluated the cytotoxic activity of the extracts, with three in vitro studies [16,49,59] (Table 1). One in vivo study [49] and two in vitro studies, using brine shrimp [17] and the Madin-Darby bovine kidney (MDBK) cell line [40], respectively, reported cytotoxic activity (Table 1). The genotoxic activity was reported in two studies for the aqueous extract (in Allium cepa bulbs and human peripheral blood) [24,28] and in one study for the ethanolic extract (in rat hepatoma cells) [21]. Conversely, three studies found no genotoxicity in ethanolic [12,25], aqueous [25], and dichloromethane [49] extracts in the in vitro model (Appendix A). Regarding mutagenicity, two studies reported no mutagenicity [21,55], while two studies also noted the presence of antimutagenic activity against cyclophosphamide in rats, which was linked to the presence of coumarin and its electrostatic interaction with DNA [56]. Toxicity was assessed in vivo with rat models [58] and an RCT [14], both showing a favorable safety profile. This RCT showed no adverse effects or changes in blood pressure and some changes in blood parameters, all within the normal range [14].

The evidence gap map for *M. glomerata* shows that there is evidence for 33 biological activities (Figure 2), with most of this evidence presenting a moderate-to-high risk of bias. In addition, there is a gap in the evidence from clinical trials.

### 2.2. Mikania laevigata

The anti-inflammatory activity of *M. laevigata* has been the most extensively studied property, with a total of nine investigations conducted [9,13,16,18,26,29,35,52,60] (Table 2). Similar to *M. glomerata*, this species did not exhibit anti-inflammatory activity in primary cultures of dystrophic skeletal muscle cells [16] (Appendix A). However, eight studies confirmed the presence of this activity. One study evaluated this activity in vitro using a screening kit to demonstrate activity against cyclooxygenase-1 and 5-lipoxygenase [18]. It was identified that six studies in vivo demonstrate the extract’s impact on intraperitoneal inflammation [9,26,52], periodontitis [13], paw edema [26,60], pleurisy [35], allergic pneumonitis [29], and pulmonary inflammation induced by coal dust in rats. Additionally, one ex vivo study focused on chick biventer cervicis and the mouse phrenic nerve-diaphragm in the context of *Philodryas olfersii* envenomation, which demonstrated protective activity against the inflammatory process [20]. The mechanism primarily associated with this activity was immunomodulation, particularly the inhibition of neutrophil [9,13], eosinophile [29], and leukocyte migration [9,26,29,60]. It was also linked to other mechanisms, including the reduced expression of TNF-α and IFN-γ [20], as well as the inhibition of prostaglandins [26].

Similar to *M. glomerata*, this species exhibits inflammatory activity against snake venom and other symptoms associated with envenomation (Table 2). Two studies reported that *M. laevigata* displayed anti-neurotoxic and anti-myotoxic activities [20,47]. These findings were attributed to the presence of coumarin [20] and tannins [47], identified as the active compounds responsible for reducing TNF-α and IFN-γ expression and for precipitating venom toxins, respectively.

Regarding bronchodilator activity, only one study was found that reported this effect in an ex vivo examination of isolated rat tracheal rings [37] (Table 2). This activity was correlated with the presence of coumarin in the extracts, with the proposed mechanism involving alterations in the cell’s ability to use or mobilize intracellular calcium, as well as the direct stimulation of calcium-activated potassium channels [37].

Antimicrobial activity was evaluated in five in vitro studies [10,30,42,45,61], three of which demonstrated that *M. laevigata* had activity against seven microorganisms [8,42,45,61] (Table 2), correlating with the presence of coumaric, cupressenic, diterpenic, and kaurenoic acids [42,61]. Consistent with the findings for *M. glomerata*, *M. laevigata* demonstrated efficacy against *S. mutans* biofilms found on children’s toothbrushes containing resistant strains of this bacterium [42] (Table 2). Furthermore, additional studies have shown that this species is effective against not only *S. mutans* biofilms but also those of *S. sobrinus* and *S. cricetus* [45]. As mentioned for *M. glomerata*, Yatsuda et al. and Lessa et al. correlated this activity with coumarin [42,61]. However, in the case of *M. laevigata*, the mouthwash had a concentration of 0.04% coumarin and an MIC of 14 µg/mL, similar to that observed with the coumarin solution.

Activity against *Streptococcus sobrinus*, *Streptococcus cricetus*, *E. faecalis*, and *Actinomyces israelii* has also been reported [42,45,61].Two studies reported no activity against *Staphylococcus aureus*, *E. coli*, *P. aeruginosa*, *Enterococcus faecalis*, *Enterococcus faecium*, and *C. albicans* [10,30]. However, one study indicated activity against *Pseudomonas aeruginosa* and *C. albicans*, primarily through the ethyl acetate fraction [45].

Regarding anticoagulant activity, an in vitro study reported that the extract of this species possessed this activity [41]. Leite et al. demonstrated that extracts acting prolongs prothrombin time and partial thromboplastin time and reduces the potential for endogenous thrombin generation, and the coumarin, terpenes, and flavonoids are correlated with this activity. However, the mechanisms of these compounds in this activity is not elucidated [41]. On the other hand, the clinical study that evaluated the blood parameters of patients who used guaco syrup did not report any change in the patients’ coagulation [14].

Antiulcerogenic activity has been demonstrated in two in vivo studies and is associated with the presence of coumarin [15,53]. Bighetti et al., based on the literature, suggest the mechanism of action by prostaglandin induction and parasympathetic secretion control [15].

This species also exhibited anthelmintic (*n* = 1) [62], antioxidant (*n* = 1) [16], anti-tumor (*n* = 1) (against tumor Hep-2 cell line) [6], and anti-germination (*n* = 1) [10] activities (Table 2). In contrast, no anti-allergic effect was observed in pleurisy in rats treated with ovalbumin [29], and there was no reproductive toxicity [38] (Appendix A).

*M. laevigata* showed no activity in cytotoxicity (*n* = 1) [6] and genotoxicity (*n* = 2 studies) [34,46] tests (Appendix A) but demonstrated antimutagenic activity in two studies—one in vitro using Salmonella cultures [31] and one in vivo in rats [46]—associated with the presence of flavonoids in the extracts, which may modify metabolic and/or detoxification rates (Table 2). Conversely, a study involving rats exposed to coal dust instillation reported an absence of this protective activity [34].

Toxicity studies conducted on *M. glomerata* were also performed on *M. laevigata*, which displayed a favorable safety profile [14] (Appendix A). This species also caused blood changes within the normal range for a healthy individual and a moderate adverse effect in relation to pyrosis [14].

The evidence gap map for *M. laevigata* shows that there is evidence for 24 biological activities (Figure 3). As was the case with *M. glomerata*, the available evidence presents a moderate-to-high risk of bias, and there is an evidence gap from clinical trials.

## 3. Discussion

To our knowledge, this is the first scoping review to systematically evaluate the available evidence on the biological activities of preparations of the leaves of *M. glomerata* and *M. laevigata*. A total of 56 primary studies were included, evaluating 38 different biological activities, of which *M. glomerata* was evaluated in relation to 33, of which 23 were positive. In the case of *M. laevigata*, a total of 24 activities were evaluated, 13 of which were positive.

A clinical trial was conducted to assess the safety of both species. The results indicated that neither posed any significant risk. Furthermore, *M. laevigata* demonstrated no evidence of cytotoxicity, genotoxicity, mutagenicity, or toxicity [6,14,31,34,37,46]. In contrast, *M. glomerata* exhibited cytotoxic effects in two in vitro studies and mutagenic effects in one. However, the observed cytotoxic effect was attributed to the low coumarin content of the plant used to prepare the extract [28].

These species have a variety of popular applications in the treatment of medical conditions, particularly being used as an anti-asthmatic, anti-inflammatory, antipyretic, antisyphilitic, antiophidic, in arthritis, rheumatism and neuralgia, as an expectorant, and in the treatment of flu and respiratory diseases in general [1]. This scoping review revealed no primary studies that had evaluated expectorant, anti-flu, antipyretic, arthritis, rheumatism, neuralgia, and antisyphilitic activity.

Only two studies explored the bronchodilator activity of these species [25,36]. The ex vivo study showed that both species have bronchodilator potential [25]; however, this effect was not observed in the RCT on asthmatic patients treated with *M. glomerata* extract [36]. The improvement observed in asthmatic patients was correlated with its anti-inflammatory potential [36].

A few studies have investigated the anti-inflammatory activity of both species, which have generally displayed such activity, with studies utilizing both species indicating that *M. laevigata* exhibits a more pronounced anti-inflammatory effect [18,26,35]. The anti-inflammatory activity was primarily attributed to coumarin in the extracts. It has been demonstrated that different coumarins exert their effects by inhibiting various pathways involved in inflammatory processes, including Toll-like receptors, the Janus kinase-signal transducer and activator of transcription, inflammasomes, Mitogen-activated protein kinase, NF-κB, and Transforming growth factor-β1 pathways [64], as well as the inhibition of pro-inflammatory enzymes like COX-2 and iNOS. Additionally, they play a crucial role in combating oxidative stress through various free-radical-scavenging mechanisms, including hydrogen atom transfer (HAT), sequential proton loss electron transfer (SPLET), and radical adduct formation (RAF) [65,66]. However, the coumarin content is higher in *M. laevigata* extracts than in *M. glomerata*, which in some cases has a content below the limit of quantification, whereas *M. laevigata* has these molecules as a chemical marker [67].

Nevertheless, the coumarin present in guaco syrups undergoes extensive first-pass metabolism, which limits its bioavailability in the plasma. Consequently, o-coumaric acid emerges as a predominant bioavailable metabolite of coumarin [68]. Two studies have examined the anti-inflammatory activity of o-coumaric acid: one investigated its effects in allergic pneumonitis and found a significant reduction in eosinophil count [29], while the other explored its role in stimulating Docosahexaenoic acid synthesis in the liver, although it concluded that coumarin’s effects were more pronounced, since o-coumaric acid did not alter the fatty acid profile [52].

The coumarins were correlated with most of the activities reported for this species, including those associated with popular use, such as an antimicrobial, anti-inflammatory, bronchodilator, and antiulcerogenic. Coumarins are well-documented for a wide range of biological activities. Molecules such as esculetin, osthole, psoralen, dicoumarol, and inophyllums are examples of coumarins with promising pharmacological properties [66]. In addition, challenges, such as low oral bioavailability, individual variations in hepatic metabolism mediated by CYP2A6, and reports of hepatotoxicity, limit the clinical advancement of new formulations [65,66]. Therefore, although promising, coumarins still require well-designed clinical trials to more robustly validate their therapeutic applications. Despite promising preclinical evidence, the translation into clinical application remains limited. For *M. laevigata* and *M. glomerata*, the studies found only hypothesize the presence of coumarins with this activity; however, these studies did not elucidate the mechanism of these compounds and confirmed this hypothesis. This lack of evidence is observed not only for the coumarins, but the other compounds correlated with the activities reported. In this way, this highlighted the importance of producing more studies focused on elucidating the mechanism of action of the compounds presented in this species.

This underscores the necessity for clinical studies that consider biological factors such as first-pass metabolism, which are pivotal for comprehending the limitations and therapeutic potential of compounds derived from these plants. A more precise evaluation of bioavailability and efficacy is imperative, particularly for compounds like coumarin and its metabolites, such as o-coumaric acid, in modulating anti-inflammatory activity. Clinically approved coumarin derivatives, such as warfarin and acenocoumarol used for the treatment of chronic venous insufficiency, highlight the translational potential of this chemical class, demonstrating that coumarin-based compounds can be successfully developed into safe and effective medications when properly studied and formulated [65,66].

In addition to coumarins, the presence of diterpenes was found to be associated with anti-inflammatory activity in *M. glomerata*. These molecules have previously been demonstrated to possess this property in the treatment of individuals with rheumatoid arthritis [69], neuroinflammation [70], and skin inflammation [71]. The chemical class in question exhibits a number of mechanisms of action, including the reduction of TNF-α, IL-1, and IL-6 expression levels in the Raw 264.7 macrophages cell line, as well as the attenuation of acetylcholinesterase and lipoxygenase activities. Additionally, there is evidence of G2/M-phase cell cycle arrest [72].

The utilization of these plants by Indigenous communities for the treatment of snakebites is also associated with this anti-inflammatory activity. Investigations into the anti-snake bite properties of these plants have demonstrated that they exert a protective effect against the inflammatory response induced by the venom of various species of *Bothrops* and *Crotalus durissus* [8,20,25,51,54]. In addition to this activity, *M. glomerata* has properties that can also be employed in the treatment of symptoms associated with snake envenomation, including analgesic and anti-hemorrhagic effects [44,51,54], and *M. laevigata* has anti-neurotoxic and anti-myotoxic effects [20,47], which protect against the paralysis caused by the venoms. However, it is important to emphasize that these plants have the potential to complement the treatment with antiophidian serum, rather than replace it.

Moreover, both species have also shown antimicrobial activity, which was predominantly associated with the presence of diterpenes [8,49,50,61]. The antimicrobial activity of terpenes, particularly diterpenes, is accomplished through diverse mechanisms of action that compromise the viability of microorganisms, as has been previously established [73,74,75]. A predominant mechanism involves the disruption of the cell membrane, whereby terpenes can compromise the integrity of the membrane, resulting in the efflux of cell contents and, consequently, the demise of the microbial cell [73,74]. In addition to the abovementioned mechanisms, some terpenes have been shown to inhibit cell wall synthesis, to impede the production of essential proteins via a process of binding to ribosomes, and to affect key enzymes for microbial metabolism [75].

Furthermore, terpenes have been demonstrated to exhibit antiviral properties by virtue of their capacity to impede crucial viral processes, including replication, protein synthesis, and viral entry [75]. However, the primary study did not establish a direct correlation between this substance and the viral activity of *M. glomerata* [40,59]. Furthermore, coumarins, which have been correlated with activity against the herpes virus [59], have also demonstrated activity against a range of other viruses, including the human immunodeficiency virus type 1, chikungunya, and hepatitis C [76]. Consequently, the development of future studies could be directed towards investigating the correlation of terpenes with this activity.

It is important to note the differences in the extracts used in the studies regarding the extraction method and the collection site. Environmental factors, such as the biome and the season in which the plant material is collected, can influence the composition of the leaves and thus the extracts obtained. The study by dos Santos et al. demonstrated this variation between collection sites, observing that extracts of *M. laevigata* leaves subjected to the same extraction process, but from the Brazilian state of Santa Catarina, had a concentration of 970 µg/mL of coumarin, while those obtained from plants collected in the state of Paraná had a concentration of 840 µg/mL [29].

In addition, the form of extraction, such as the use of dried or fresh leaves, and the type of solvent used, can alter the composition of the final extract. The same study by dos Santos et al. showed that extracts obtained with ethanolic solvents for *M. laevigata* and *M. glomerata* had a higher concentration of coumarin than aqueous extracts using the same plant sample [29]. Another study showed that extracts from fresh *M. glomerata* leaves contained only traces of coumarin, while extracts from dried leaves ranged from 254 to 2794 µg/mL [77].

On the other hand, the extracts used in the clinical studies by Bertol et al. showed lower concentrations of coumarin than those observed in the studies, with 4 µg/mL for *M. glomerata* and 120 µg/mL for *M. laevigata* [14]. These data highlight the importance of the detailed characterization of the extracts used in studies, as extracts with different compositions may have different biological activities. However, most of the studies included in this review did not characterize the extracts used, which makes it difficult to correlate these studies with clinical practice since the exact composition of the extracts used is unknown.

This lack of characterization makes translation into clinical practice difficult and also makes it difficult to understand the mechanisms of action of the biological activities observed. Many of the included studies only present hypotheses about the possible mechanisms of the observed activities. In addition, this review showed that most in vivo studies have a high risk of bias, often due to the lack of randomization and blinding of the investigators. This highlights the need not only for clinical trials, but also for better designed in vivo studies with robust methodologies, including proper randomization and investigator blinding, to ensure reliable and unbiased results in future research to better elucidate the activities of these extracts.

It is recommended that future preclinical and clinical studies of preparations using *M. laevigata* and *M. glomerata* be well chemically characterized and evaluate their stability, as this represents a critical factor for ensuring safety and efficacy, since the degradation of products may interfere with the outcomes of clinical studies.

This study has several limitations. Using distinct extraction methodologies and solvents may potentially influence the extracted compounds and, consequently, the observed biological activities. Most studies included were pre-clinical, while the two RCTs incorporated were limited to the Brazilian population. The risk-of-bias analysis demonstrated that the majority of studies conducted on animal models exhibited a moderate-to-high risk of bias.

## 4. Materials and Methods

This study was conducted in accordance with the recommendations of the Joanna Briggs Institute [78] and reported by following the Preferred Reporting Items for Systematic Reviews and Meta-Analyses Extension for Scoping Reviews (PRISMA-ScR) Checklist [79]. The study protocol was registered on the Open Science Framework (DOI: 10.17605/OSF.IO/AZN5T). Two authors were responsible for conducting all stages of the study selection, data extraction, and quality assessment independently. In the event of any discrepancies, a third author was consulted.

### 4.1. Literature Selection

A systematic search was conducted in the PubMed, Scopus, and Web of Science electronic databases on 7 October 2024. No time or language restrictions were applied; the complete search strategy is available in Appendix A. A manual search was also conducted on the references cited in the included studies.

This scoping review includes the primary studies to answer the review question, “What are the biological activities of extracts derived from the leaves of *M. glomerata* and *M. laevigata*?”. In the initial selection phase, which included reading titles and abstracts, articles were excluded if they met any of the following criteria: (1) extracts were from both species combined; (2) extracts were from other parts rather than leaves; (3) articles were not primary studies (e.g., systematic reviews); and (4) articles were written in non-Roman characters. In the subsequent eligibility phase, articles were excluded if they did not meet the criteria established. The screening and eligibility step was carried out using Rayyan online software (Rayyan Systems, Inc., Cambridge, MA, USA, free 892 version).

### 4.2. Data Extraction and Quality Assessment

Data extraction included study metadata (e.g., authors, year of publication), information about the extract used (e.g., plant used, plant collection site, extraction method, extraction solvent), and details of the study conducted (study design, models used, results). In cases where the evaluated activity was present, the possible mechanisms of action and molecules associated with this activity were extracted.

For the in vivo studies, the risk of bias was assessed using the SYRCLE tool, which includes ten different domains that assess selection (sequence generation, baseline characteristics, and allocation blinding), performance (randomization and concealment), detection friction (random assessment of outcomes and concealment), communication (incomplete outcome data and selective communication of outcomes) and other sources of bias [80]. The domains were rated as ‘low’, ‘unclear’, or ‘high’ risk of bias.

For randomized clinical trials, the RoB2.0 was used [81]. This tool assesses trials in five main domains: selection bias, performance, detection, attrition, and communication. The domains were rated as ‘low risk of bias’, ‘some concern’, or ‘high risk of bias’.

### 4.3. Data Synthesis

A narrative summary of the characteristics of the included studies, organized by species, characteristics of the studies conducted, and biological activities, is presented in the tables. Additionally, an evidence gap map was constructed using data on the risk of bias of the included studies, biological activities assessed, and species with R/Rstudio 4.4.2 software.

## 5. Conclusions

The widespread and often indiscriminate use of *Mikania laevigata* and *Mikania glomerata* without robust scientific evidence raises significant concerns regarding their safety and efficacy. The lack of standardized protocols for extract preparation, frequent absence of chemical characterization, and wide variation in methods used to obtain the extracts results in inconsistent pharmacological effects or even potential toxicity. Furthermore, in many countries, including Brazil, herbal medicines may be marketed with less stringent regulatory oversight compared to conventional drugs, increasing the risk of exposing the population to ineffective or harmful products.

This review highlights the potential biological activities of *M. laevigata* and *M. glomerata* extracts, particularly their anti-inflammatory properties. However, discrepancies in anti-inflammatory potential between the two species, divergence in their chemical composition, and interchangeable use by the population underscore the need to elucidate the distinctions between their biological activities. In addition, gaps in evidence for certain biological activities and the lack of chemical characterization make it difficult to elucidate their mechanisms of action, limiting the translation of findings into clinical practice. Despite significant popular use of these species, scientific evidence remains limited, highlighting the importance of conducting new studies with more rigorous designs to clarify their true therapeutic potential and establish clear regulatory standards for the safe and effective use of phytotherapeutic agents derived from these plants.

## Figures and Tables

**Figure 1 pharmaceuticals-18-00552-f001:**
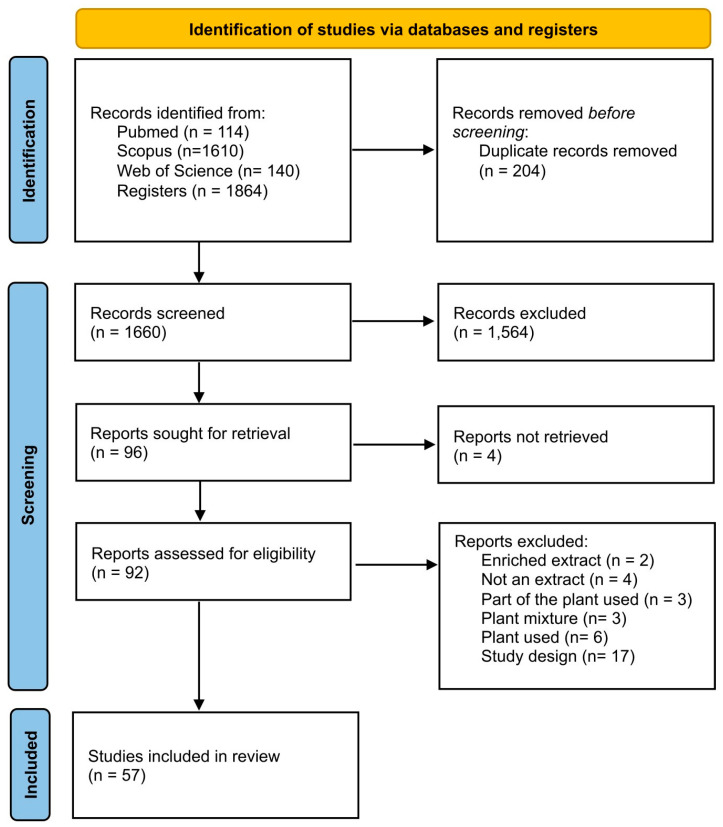
PRISMA 2020 flow diagram.

**Figure 2 pharmaceuticals-18-00552-f002:**
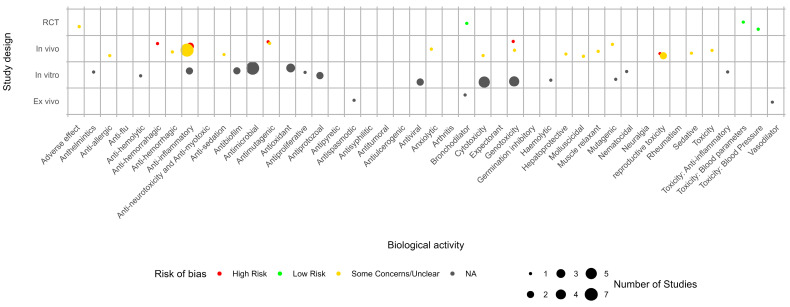
Evidence gap map for *Mikania glomerata*. NA: not applicable.

**Figure 3 pharmaceuticals-18-00552-f003:**
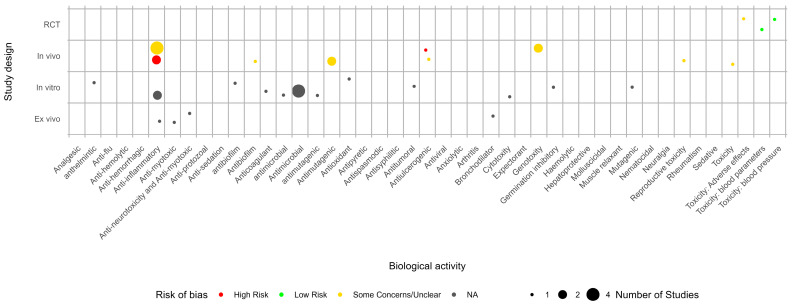
Evidence gap map for *Mikania laevigata*. NA: not applicable.

**Table 1 pharmaceuticals-18-00552-t001:** *Mikania glomerata* reported activity.

Biological Activity	Method	Model	Solvent	Related Compound	Probable Mechanism of Action	Ref.
Antispasmodic	Ex vivo	Guinea pig ileum and rat jejunum	EtOH 96% and 50%	NR	Reduced acetylcholine and histamine levels	[8]
Analgesic	In vivo	Mice on *Bothropoides jararaca* envenomation	Water	NR	Reduction in contortions	[54]
Anthelmintics	In vitro	*Toxocara canis* and *Ancylostoma caninum*	EtOH 96%	Diterpenes	Prevents egg embryogenesis	[62]
Anti-allergic	In vivo	Pleurisy in rats caused by ovalabulmin	EtOH and fraction of DCM	NR	Inhibition of granulocyte infiltration following antigen challenge; inhibition of antigen-induced mast cell degranulation and partial inhibition of PAF-induced granulocyte infiltration	[32]
Antimicrobial	In vitro	Culture of *Pseudomonas aeruginosa*, *Salmonella typhimurium*, *Klebsiella pneumoniae*, *Bacillus cereus and Escherichia coli*	MeOH	NR	NR	[17]
		Culture of *Lactobacillus casei*, *S. sanguinis*, *S. mutans*, *E. faecalis*, *S. salivarius*, *S. mitis*, *Streptococcus sobrinus*	Dichloromethane	Diterpene	NR	[49]
		Culture of *Actinomyces naeslundii*, *Aggregatibacter actinomycetemcomitans*, *Enterococcus faecalis*, *Fusobacterium nucleatum*, *Porphyromonas gingivalis*, *Prevotella intermedia*, *P. nigrescens*, *P. melaninogenica*, *Propionibacterium acnes*	DCM	Diterpenes	NR	[50]
		Culture of *Streptococcus mutans*, *Streptococcus sobrinus*, *Streptococcus cricetus*	EtOH 70%	Coumaric, cupressenic, diterpenic and kaurenoic acids	Inhibition of growth and inhibition of sucrose-dependent adherence of mutans streptococci cells to a glass surface at sub-MIC levels	[61]
		Culture of *Streptococcus mutans*	EtOH	Coumarin	NR	[42]
Antibiofilm	In vitro	*Streptococcus mutans* biofilm	DCM	Diterpenes	NR	[49]
			EtOH	Coumarin	NR	[42]
Anti-hemorrhagic	In vivo	Mice on *Bothrops jararacussu*, *Bothrops moojeni*, *Bothrops alternatus*, *Bothrops neuwiedi* envenomation	EtOH	NR	NR	[44]
		Wistars rats on *Bothropoides jararaca* envenomation	EtOH 70%	NR	Inhibition of metalloproteinases	[51]
Anti-inflammatory	In vitro	Peripheral blood mononuclear cells	EtOH 55%	Flavonoids	Blocking the stimulation of lymphocytes	[48]
		Screening kits from Cayman Chemical’s ACE	EtOH 70%	NR	Dual inhibition of cyclooxygenase-1 and 5-lipoxygenase	[18]
	In vivo	Edema in mice caused by *Bothropoides jararaca* envenomation	Water and EtOH	NR	NR	[25]
		Pleural oedema in rats caused by biogenic, amines, carrageenan	EtOH and a fraction of DCM	NR	Inhibited leukocyte infiltration	[32]
		Edema in Wistar rats caused by *Crotalus durissus* venom	Water	NR	NR	[33]
		Edema of Wistars rats on *Bothropoides jararaca* envenomation	EtOH 70%	Coumarin	Inhibition of inflammatory toxins (phospholipase A2)	[51]
		Mice on *Bothropoides jararaca* envenomation	Water	NR	Inhibition of inflammatory toxins (phospholipase A2)	[54]
		Paw edema, pleurisy, peritoneal inflammation in Wistar rats	Water	Coumarin	Inhibition of prostaglandin synthesis	[26]
		Pulmonary inflammation caused by coal dust in Wistar rats	EtOH 70%	NR	Protective role in the oxidation of thiol groups	[35]
Antimutagenic	In vivo	Swiss albino mice treated with cyclophosphamide	EtOH 70%	Coumarin	Interact electrostatically with the DNA molecule, so it can interfere with the action of cyclophosphamide	[56]
Antioxidant	In vitro	DPPH assay	EtOH 70%	NR	NR	[57]
		DPPH assay	Tea	NR	NR	[19]
		Oxygen Radical Absorbance Capacity, DPPH, and oxidative stress in dystrophic primary muscle cells of mice	EtOH 70%	Phenolic and caffeoylquinic acids	NR	[16]
Antiproliferative	In vitro	*Allium* cepa bulb	water	Coumarin	Apoptotic mechanisms that are likely to have been activated at higher concentrations	[24]
Antiprotozoal	In vitro	*Leishmania amazonensis*	EtOH 90%	NR	NR	[43]
Anti-sedation	In vivo	Wistar rats on *Crotalus durissus* envenomation	Water	NR	NR	[33]
Antiviral	In vitro	*Suid alphaherpesvirus 1* and *Bovine Herpesvirus type-1*	Water	NR	NR	[40]
		Human *herpes simplex virus type 1* and *2*	EtOH 70%	Coumarin *	NR	[59]
Anxiolytic	In vivo	Mice in light/dark box test	EtOH 70%	NR	Increases GABA levels and decreases glutamate and aspartate concentrations in the hippocampus of mice	[57]
Bronchodilator	Ex vivo	Human bronchi and guinea-pig trachea	Water and EtOH	coumarin	Reduced histamine-induced contraction and induced a depression of the maximal response	[25]
Cytotoxicity	In vitro	MDBK cell line	Water	NR	NR	[40]
	In vitro	Brine shrimp	MeOH	NR	NR	[17]
Genotoxic	In vitro	Rat hepatoma cells	EtOH 80%	NR	NR	[21]
		*Allium* cepa bulb	Water	Coumarin	Apoptotic mechanisms that are likely to have been activated at higher concentrations	[24]
		Human peripheral blood	Water	Coumarin **	NR	[28]
Haemolytic	In vitro	Sheep blood	EtOH 90%	NR	NR	[43]
Hepato-protective	In vivo	Mice treated with carbon tetrachloride	Tea	inulin-type fructan	Protect the liver from carbon tetrachloride-induced hepatotoxicity	[19]
Molluscicidal	In vivo	*Subulina octona*	Water	saponins, tannins, and flavonoids	Saponins lyse the cell and complex with steroids. Tannins complex proteins and cause them to precipitate. Flavonoids modify the cytochrome P450 enzyme	[26]
Nematocidal	In vitro	*Pratylenchus jaehni* and *Pratylenchus zeae*	EtOH	NR	NR	[63]
Vasodilatador	Ex vivo	Isolated rat superior mesenteric vascular bed and rat aorta	Water and EtOH	NR	NR	[25]

* For human herpes simplex virus type 1; ** in this case, the lack of it; DCM, dichloromethane; DPPH, 2,2-Diphenyl-1-picrylhydrazyl radical; EtOH, ethanol; MeOH, methanol; NR, not reported.

**Table 2 pharmaceuticals-18-00552-t002:** *Mikania leavigata* reported activity.

Biological Activity	Method	Model	Solvent	Related Compound	Probable Mechanism of Action	Ref.
Germination inhibitory	In vitro	*Lactuca sativa* L. seed, Boston White variety	EtOH 96%	NR	Allelopathic activity	[10]
Anthelmintic	In vitro	*Toxocara canis* and *Ancylostoma caninum*	EtOH 96%	Diterpenes	Prevents embryogenesis of *T. canis* eggs	[62]
Antibiofilm	In vitro	Culture of *Streptococcus mutans*, *Streptococcus sobrinus*, *Streptococcus cricetus*	HEX, EtOAc, and n-BU: water (1:1, *v*/*v*)	NR	NR	[45]
		Culture of *Streptococcus mutans*	EtOH	Coumarin	NR	[42]
Anticoagulant	In vitro	Human plasma	EtOH	Coumarin, diterpenes, flavonoids, phenylpropanoids	Prolongs prothrombin time and partial thromboplastin time and reduces the potential for endogenous thrombin generation	[41]
Anti-inflammatory	In vitro	Screening kits from Cayman Chemical’s ACE	EtOH 70%	Hispidulin, 5-O-E-caffeoylquinic acid, 3-O-E-cumaroylquinic acid, cumaric acid, 3,4-di-O-E-caffeoylquinic acid, apigenin	Dual inhibition of cyclooxygenase ASE-1 and 5-lipoxygenase	[18]
	In vivo	Swiss mice treated with intraperitoneal injection of carrageenan	EtOH 70%	Coumarin	Inhibit neutrophil migration by various means, such as leukocyte-endothelium interaction (rolling and adhesion) or neutrophil transmigration, and also by decreasing vascular permeability	[9]
		Wistar rat periodontitis	EtOH 65%	NR	Reduced RANKL expression and neutrophil migration	[13]
		BALB/c mice sensitized via intraperitoneal injection with ovalbumin and aluminum oxide	EtOH water 1:2	Coumarin and o-coumaric acid	Stimulation of Docosahexaenoic acid synthesis in the liver	[52]
		BALB/c mice with allergic pneumonitis sensitized with ovalbumin and aluminum oxide	EtOH-water 1:2 and water	Coumarin and o-coumaric acid	Inhibition of leukocyte and eosinophil influx	[29]
		Carrageenin-induced paw oedema in Wistar rats	Water	Diterpenes	Reduced polymorphonuclear leukocyte migration	[60]
Anti-inflammatory	In vivo	Paw edema, pleurisy, peritoneal inflammation in Wistar rats	Water	Coumarin	Inhibition of prostaglandin synthesis and decreased leukocyte migration	[26]
		Pulmonary inflammation caused by coal dust in Wistar rats	EtOH 70%	NR	Protective role in the oxidation of thiol groups	[34]
	Ex vivo	Chick biventer cervicis and Mouse phrenic nerve-diaphragm on Philodryas olfersii envenomation	EtOH	Coumarin	The expression of TNFa and IFNy has been attenuated	[20]
Antimicrobial	In vitro	Culture of *E. faecalis*, *P. aeruginosa*, *A. israelii*, and *C. albicans*	HEX, EtOAc, and n-BU: Water (1:1, *v*/*v*)	NR	NR	[45]
		Culture of *S. mutans*, *S. sobrinus*, *S. cricetus*	EtOH 70%	Coumaric, cupressenic, diterpenic, and kaurenoic acids	Inhibition of growth and inhibition of sucrose-dependent adherence of mutans streptococci cells to a glass surface at sub-MIC levels	[61]
		Culture of *S. mutans*	EtOH	Coumarin	NR	[42]
Antimutagenic	In vitro	TA98, TA97a, TA100, and TA1535 strains of *Salmonella typhimurium.*	Water	Flavonoids	Cytochrome P450 system blockade	[34]
	In vivo	Mice with genotoxicity induced by methyl methanesulfonate and cyclophosphamide	EtOH 70%	Flavonoids	Alter the rate of metabolism and/or detoxification	[38]
Anti-neurotoxicity and Anti-myotoxic	Ex vivo	Chick biventer cervicis and Mouse phrenic nerve-diaphragm on *Philodryas olfersii* envenomation	EtOH	Coumarin	Eversible neuromuscular blockade and attenuated expression of TNFa and IFNy	[20]
		Mouse phrenic nerve-diaphragm muscle on *Crotalus durissus terrificus* or *Bothrops jararacussu* envenomation	EtOH 50%	Tannic acid	Formation of precipitates with toxins	[47]
Antioxidant	In vitro	Oxygen Radical Absorbance Capacity, DPPH and oxidative stress in dystrophic primary muscle cells of mice	EtOH 70%	Phenolic and caffeoylquinic acids	NR	[16]
Antitumoral	In vitro	Tumor Hep-2, HeLa cell line	HEX, EtOAc, CHL, and EtOH 50%	Coumarin, phenolic compounds	Inhibit their proliferation	[6]
Antiulcerogenic	In vivo	Wistar rats with Indomethacin-induced ulcer	EtOH 70%	Coumarin	Induce prostaglandin production and influence parasympathetic secretion control	[15]
		Indomethacin-induced gastric lesion in mice	EtOH 80%	Coumarin and mucilage	NR	[53]
Bronchodilator	Ex vivo	Isolated rat tracheal rings	EtOH 70%	Coumarin	Changes in the cell’s ability to use or mobilize intracellular calcium and direct stimulation of calcium-activated potassium channels	[37]

CHL, chloroform; DPPH, 2,2-Diphenyl-1-picrylhydrazyl radical; EtOH, ethanol; EtOAc, ethyl acetate; HEX, Hexane; n-BU, n-butanol; NR, not reported.

## Data Availability

The data are availably in the article or in the Appendix A.

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
