# Peer review of "Biological Activities of Mikania glomerata and Mikania laevigata: A Scoping Review and Evidence Gap Mapping"

_pharmaceuticals, 2025, doi:10.3390/ph18040552_

Round 1

Reviewer 1 Report

Comments and Suggestions for Authors

This study provides a comprehensive scoping review of the biological activities of Mikania glomerata and Mikania laevigata, commonly known as guaco. By analyzing 57 studies, the authors highlight the strong preclinical evidence for their anti-inflammatory properties, particularly in respiratory conditions. However, the manuscript also reveals significant evidence gaps regarding other traditional uses, such as their expectorant and antipyretic effects. The article underscores the need for further clinical studies to validate these popular medicinal claims.

My first impression is positive. This type of article is always very useful for the scientific audience. However, I have a few points to point:

  1. In the abstract please remove the conclusion and the results part. Just make sure that briefly is explained what is the manuscript about.
  2. Please insert Mikania glomerata, Mikania laevigata as a key words.
  3. In the discussion part, the discussion is very superficial and brief. The information is given as a kind of reference data. Please comment on the information in detail and write down your critical comments. Only then, it will be valuable and interesting for the reader.
  4. The conclusion too. Please make a conclusion about the material you are conveying, not what properties you have read that it has and describe them one by one.
  5. Please revise the review article by delving into details and presenting the information coherently and consistently. The way it is currently presented shows that you have very superficially extracted the information from the references.

Based on the above. I recommend that the article be reconsidered after a major revision.

Reviewer 2 Report

Comments and Suggestions for Authors

The manuscript presents a comprehensive scoping review on the biological activities of Mikania glomerata and Mikania laevigata, highlighting their anti-inflammatory potential and diverse pharmacological applications. The study effectively maps existing evidence and identifies research gaps, offering valuable insights for future clinical trials. Strengths include a well-structured methodology, systematic risk of bias assessment, and clear presentation of results. However, the manuscript has some weaknesses, such as the predominance of preclinical studies, limited clinical trials, and moderate to high bias in animal studies. Additionally, the discussion could benefit from a more in-depth exploration of the mechanistic pathways underlying the observed activities. Overall, the work is well-conceived and offers significant contributions to the field, though further clinical investigations are essential to confirm therapeutic efficacy. I recommend acceptance after minor revisions.

Here are some recommendations:

The authors are invited to provide a deeper discussion of methodological limitations, particularly regarding studies with moderate to high risk of bias, and suggest improvements like standardized protocols and proper randomization.

The manuscript needs more detailed exploration of the biochemical pathways involved, especially concerning coumarins and diterpenes, to strengthen the understanding of observed effects.

The authors could better connect preclinical findings to potential clinical applications, emphasizing the need for human trials to confirm efficacy and safety.

The authors are invited to discuss the impact of different extraction methods, solvents, and plant origins on the chemical composition and biological activities of the extracts.

Comments on the Quality of English Language

The quality of English is quite good, clear, concise, and scientifically appropriate. However, a few minor improvements could enhance readability.

Round 2

Reviewer 1 Report

Comments and Suggestions for Authors

After reviewing the revised version I do not find any other points to raise, Now the manuscript looks much better.